



# The Real Challenges for Climate and Weather Modelling on its Way to Sustained Exascale Performance: A Case Study using ICON (v2.6.6)

Panagiotis Adamidis[1], Erik Pfister[1], Hendryk Bockelmann[1], Dominik Zobel[1], Jens-Olaf Beismann[2], and Marek Jacob[3]

[1]Application Support, German Climate Computing Centre (DKRZ), Bundesstraße 45a, 20146 Hamburg, Germany
[2]NEC Deutschland GmbH, Fritz-Vomfelde-Straße 14, 40547 Düsseldorf, Germany
[3]Research and Development, Deutscher Wetterdienst (DWD), Frankfurter Straße 135, 63067 Offenbach, Germany

**Correspondence:** Panagiotis Adamidis (adamidis@dkrz.de) and Erik Pfister (pfister@dkrz.de)

**Abstract.** The weather and climate model ICON (ICOsahedral Nonhydrostatic) is being used in high resolution climate simulations, in order to resolve small-scale physical processes. The envisaged performance for this task is 1 simulated year per day for a coupled atmosphere-ocean setup at global 1.2 km resolution. The necessary computing power for such simulations can only be found on exascale supercomputing systems. The main question we try to answer in this article is where to find

sustained exascale performance, i. e. which hardware (processor type) is best suited for the weather and climate model ICON and consequently how this performance can be exploited by the model, i. e. what changes are required in ICON's software design so as to utilize exascale platforms efficiently. To this end, we present an overview of the available hardware technologies and a quantitative analysis of the key performance indicators of the ICON model on several architectures. It becomes clear that domain decomposition-based parallelization has reached the scaling limits, leading us to conclude that the performance of

a single node is crucial to achieve both better performance and better energy efficiency. Furthermore, based on the computational intensity of the examined kernels of the model it is shown that architectures with higher memory throughput are better suited than those with high computational peak performance. From a software engineering perspective, a redesign of ICON from a monolithic to a modular approach is required to address the complexity caused by hardware heterogeneity and new programming models to make ICON suitable for running on such machines.

## 1 Introduction

High-performance computing in the early 2020s is reaching a new era with the availability of the first exascale systems for scientific simulations (e.g. the first official LINPACK (LINear system PACKage) exascale System Frontier, see Dongarra and Geist (2022), or the first planned European Exascale HPC (High Performance Computing) System JUPITER in Jülich). These computer systems will enable unprecedented accuracy in climate research. For example, it will be possible to calculate

ensembles of climate processes over several decades and on spatial scales of 1 km globally (Hohenegger et al., 2023). Such kilometre-scale climate models offer the potential to transform both science and its application, eventually leading to the





creation of digital twins of the Earth (Hoffmann et al., 2023). However, this technology poses not only a programming challenge for climate science, namely the development of adapted seamless simulation systems, but it must also be ensured that the enormous power consumption of these machines can be utilised efficiently (Bauer et al., 2021).

The upcoming exascale supercomputers are massively parallel processing systems. They consist of several thousands of nodes, whereby certainly not only x86 architectures will be used, but also other architectures such as GPU, Vector or ARM have to be considered, as they all make it to the current Top500 list[1], e. g. the Supercomputer Fugaku of the RIKEN Center for Computational Science with the A64FX architecture or the Earth Simulator SX-Aurora TSUBASA which uses Vector Engines.

  The task of efficiently using exascale systems is already the subject of various research (e. g. in the Exascale Computing
Project, see Messina (2017)). Besides a good scaling behaviour, which is given in ICON (Giorgetta et al., 2022), an optimal utilization of the certain processing units is necessary in order to run kilometre-scale climate simulations with acceptable performance. In section 2 we present a survey of the available hardware technologies and outline software aspects of the ICON model (Giorgetta et al., 2018; Crueger et al., 2018; Zängl et al., 2015). Furthermore, by specifying the model configuration and experiment, we determine the scope of our investigations. The multi-node scalability is explored in section 3 whereas in
section 4 an assessment of the single-node performance is given, as we consider this the key to sustained exascale performance. Section 5 highlights the importance of energy efficiency as energy consumption becomes a critical cost factor.

## 2 Exascale in Climate Science?

### 2.1 Hardware Perspective

Taking a look at the trend of the top 10 HPC systems in the Top500 list, it becomes clear that CPUs (Central Processing Units)
alone are no longer sufficient to equip an exascale system. GPU (Graphics Processing Unit) accelerators have dominated the Top500 list since 2015 at the latest and are currently the centrepiece of all (pre-) exascale systems. However, since many national weather services in particular used vector processors, and in some cases still do, the NEC SX-Aurora Vector Engine will also be the focus of the investigations.

---

[1]Top500 list:https://www.top500.org/





**Table 1.** Different architectures and some of their specialities.

| Architecture | Specialties |
|---|---|
| CPUs | • General-purpose computing capabilities for a wide range of workloads<br>• Mature software ecosystem and a wide range of programming languages and tools available<br>• Large memory capacity and bandwidth |
| GPUs | • Extremely high parallel compute power<br>• Requires special compilers and language extensions<br>• Is used in combination with CPUs in a heterogeneous computing architecture<br>• Supports HBM2 memory for fast data transfer rates |
| NEC Aurora | • High memory bandwidth<br>• Supports HBM2 memory for fast data transfer rates<br>• Supports a variety of programming languages and tools<br>• Requires vendor-specific compiler<br>• Can be used as main processor or as an accelerator |

It is important to note that the sustained performance of each architecture depends on the specific workload being executed
and the respective implementation of the tasks. Also note that the most effective architecture depends on the particular re-
quirements of each application. In general, however, GPUs are often well-suited for highly parallel workloads such as machine
learning, while CPUs may be more appropriate for general-purpose computing and applications with more irregular data ac-
cess patterns. Specialized architectures such as NEC SX-Aurora TSUBASA may be optimal for specific types of scientific
computing workloads (e.g. bandwidth-limited applications).

We use nodes equipped with AMD EPYC 7763, NVIDIA A100 SXM4, NEC SX-Aurora TSUBASA VE10AE, or NEC SX-
Aurora TSUBASA VE30A as representative models for the hardware architectures mentioned above in Table 1. The hardware
characteristics of the different nodes and processors are listed in Table 2. The GPUs and vector engines are integrated into
CPU host systems. These CPU systems manage the respective accelerator. The CPU compute capability could in theory be
used together with the accelerator in a heterogeneous fashion. Such heterogeneous computing complicates the programming
model and, in the context of ICON, this was considered not worth it as the theoretical compute performance of the accelerator
is much higher than that of the host. However, the host capabilities can be utilized in ICON by assigning output processes to
the host's CPU. Furthermore, the host is used for reading and processing of the initial state and setting up the model. In this
study, we focus on the sustained performance of the time integration loop and ignore the initial phase and output so that we
compare only the accelerator performance.





**Table 2.** Hardware characteristics. Theoretical maximum performance metrics of the compared nodes are for double precision number for a full node.

| Node configuration | 2x AMD EPYC 7763 | 4x NVIDIA A100 SXM4 | 8x NEC SX-Aurora TSUBASA | |
|---|---|---|---|---|
| | | | VE10AE | VE30A |
| Architecture | CPU | GPU | Vector Engine | Vector Engine |
| Host CPU | — | 2x AMD 7713 | AMD 7402P | AMD 7443P |
| Number of cores | 128 | 13824 | 64 | 128 |
| Core base clock speed [GHz] | 2.450 | 1.065 | 1.584 | 1.600 |
| Max. clock speed [GHz] | 3.500 | 1.410 | 1.584 | 1.600 |
| Theor. max. Flops [GFLOP/s] | 5018 | 38800 | 19464 | 39322 |
| Type of memory | DDR4 | HBM2E | HBM2 | HBM2E |
| Memory capacity [GB] | 512 | 320 | 384 | 768 |
| Theor. max. memory bandwith [GB/s] | 410 | 8156 | 10800 | 19600 |
| Node interconnect | InfiniBand HDR100G | InfiniBand HDR100G | InfiniBand HDR100 | InfiniBand NDR200 |
| Launch date | March 2021 | June 2021 | February 2018 | July 2023 |

## 2.2 Software Perspective

Hardware alone does not deliver performance. The software design of the model must be adapted to the individual hardware of each platform in order to fully exploit its performance. A major challenge that comes with heterogeneous hardware is the variety of different programming models, which might be used to enable the model to run on the various processing units (Fang et al., 2020).

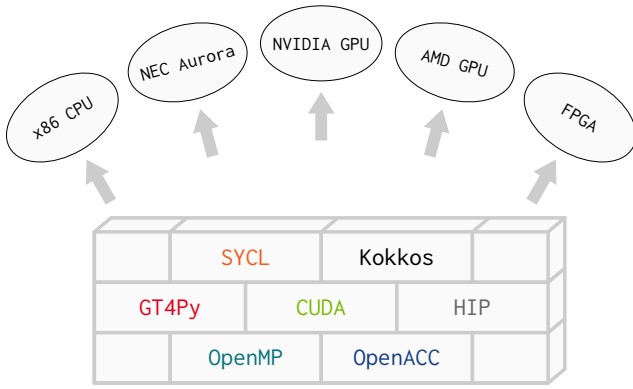

**Figure 1.** The wall of programming models for targeting heterogeneous architectures.



Although a vendor-specific solution will be very efficient, as NVIDIA reports in Fuhrer (2023), it is tailored to the specific architecture and is not portable. A community model like ICON is required to run on different architectures. For this reason, performance portability across platforms is crucial and choosing the appropriate programming model becomes a difficult task as not all of them support all types of accelerators (Figure 1). Furthermore, ICON is based on FORTRAN, which limits the use of possible programming models to directive-based models such as OpenACC or OpenMP. Applying programming models

with higher levels of abstraction and therefore higher performance portability, such as Kokkos (Trott et al., 2022) or SYCL (Rovatsou et al., 2023), would require a complete rewrite of the code.

### 2.2.1  ICONs monolithic code base

The current parallel programming model in ICON is diverse. Support for distributed-memory parallel systems in ICON has been implemented using the Message Passing Interface (MPI, see Message Passing Interface Forum (2021)). Multiple ICON

processes run concurrently on multiple nodes and each process is assigned a portion of the horizontal domain or domains if online-nesting is used (see Section 3). The boundary information required to solve the differential equations for each grid point in each local domain is exchanged between the processes via MPI messages over the network. Besides the classical domain-decomposition via MPI, ICON supports three additional programming models for parallelizing the processes themselves. These are shared memory parallelization with OpenMP, automatic or semi-automatic vectorisation enabled by the compiler,

and OpenACC for accelerator devices with discrete memory.

To take advantage of multi-core systems with shared memory, the OpenMP programming model has been implemented to enhance the computational performance of each MPI process. OpenMP is primarily being used to compute all time-dependent routines in a thread-parallel manner. In addition, vendor-specific pragmas have been added to the code to guide certain compilers, such as the NEC compiler, to the most efficient vectorisation of individual loops. Most of the pragmas mark loop iterations

as independent of each other, even though the compiler has initially noticed that a code structure, like index lists, could theoretically imply a loop dependency. Recent efforts have introduced another programming model to ICON to take advantage of the massively parallel computing power of accelerators such as GPUs (Giorgetta et al., 2022). The OpenACC Application Programming Interface (OpenACC API) was chosen in this regard as it was the only practical solution to stay close to the original Fortran code. OpenACC is used to manage the discrete memory of the accelerator and also to parallelize the ICON

computations. However, loops are parallelized at a lower level in the ICON call tree with OpenACC compared to OpenMP.

The parallelization methods are not mutually exclusive. A hybrid approach of MPI plus parallelisation within each process is possible. Compiler-assisted vectorisation and OpenMP can also be combined. When combining MPI and OpenACC, data exchange messages can be sent and received directly from the dedicated accelerator memory without the need to copy the data to the host memory first. This requires an accelerator-aware implementation of MPI. For small problem sizes and testing

purposes, ICON can also be run without MPI. Just OpenMP and OpenACC are currently mutually exclusive in the ICON code, and the OpenMP target offloading as defined in the 4.5 and later standards is not supported in the main code. However, linked libraries could in principle be compiled e.g. using OpenMP and they can be linked to an OpenACC accelerated binary. Furthermore, different ICON binaries compiled with different process-specific parallelization methods can be combined using





MPI, as long as all processes use the same MPI library. The reader is referred to chapter 8 of Prill et al. (2023) for more
information on the ICON parallelization.

ICON's software design so far takes a monolithic approach. All of the above parallel programming methods have been implemented in the same source code and the distinction between them is made by `#if` and `#ifdef` macros and other directives. Although ICON functionalities are separated in modules and imported when needed, the extensive use of rather complex derived data structures throughout the code mitigates some of the advantages from encapsulation. Some code is
specifically optimized for certain architectures, guarded by preprocessor macros and augmented with directives (cf. Table 3). Apart from the different directives (OpenACC, OpenMP, NEC-Aurora), the loops for the different architectures are also written in different variants in order to optimally utilize the different processing units. However, this bloats the code and makes it even more difficult to adapt the model to new architectures. The later is a hard requirement for ICON, since it is meant to be a community tool and shall be able to run on all upcoming supercomputers.

**Table 3.** Amount of directives used for different architectures and conditions using them in ICON.

|  | NEC directives | OpenMP directives | OpenACC directives |
| --- | --- | --- | --- |
| Amount of directives | 800 | 5750 | 15100 |
| Macro conditions using them | 200 | 150 | 75 |

To prepare the model for the exascale era of supercomputing systems, ICON is currently undergoing a major refactoring. Given the heterogeneous hardware, performance portability is crucial. For this purpose, the code base is converted from a monolithic code into a modularized, scalable and flexible code (cf. Figure 2).

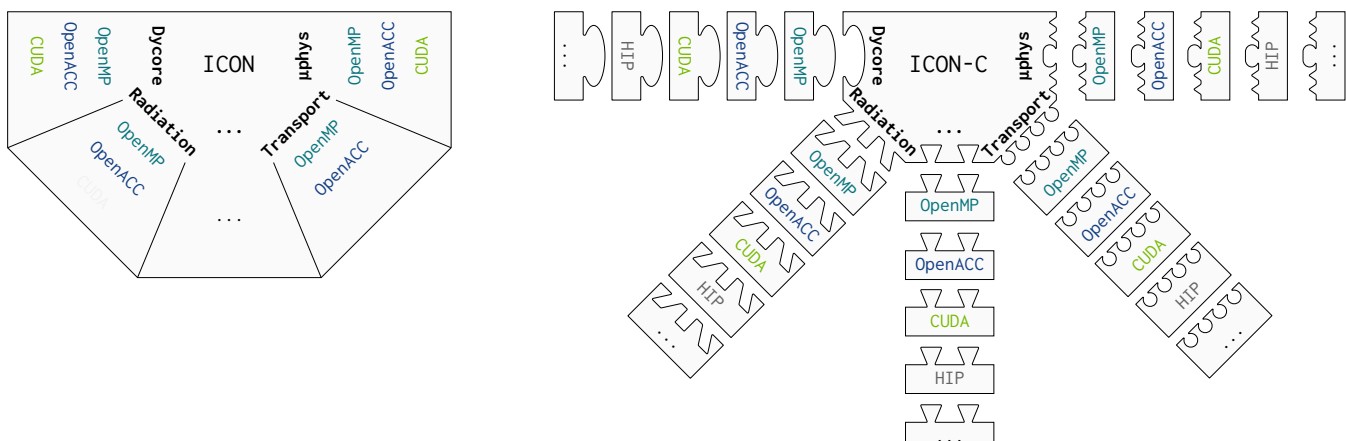

**Figure 2.** Left hand side: Visualization of ICON's monolithic software design. Right hand side: Visualization of the new ICON-Consolidated (ICON-C) software design. The model consists of multiple encapsulated modules and each module can be independently ported to new architectures, using different programming paradigms. A well defined interface should integrate the individual modules together into the main time loop.





### 2.2.2 ICON configuration

A specific model configuration had to be chosen to study the sustained performance of ICON. From the available options
we chose a configuration based on the Deutscher Wetterdienst's (DWD) operational setup for numerical weather prediction
(NWP). All components of ICON that are used in this configuration have been ported in a joined effort by MeteoSwiss, its
partners and DWD to GPU using the OpenACC API (Prill et al., 2023; Osuna and Consortium for Small-scale Modeling, 2023).
These components have also been optimized for the NEC Aurora, as that is DWD's operational machine. The support for CPUs
of this configuration is provided almost naturally, as the CPU mode of ICON is the foundational Fortran implementation and
as the predecessor of DWD's current machine was CPU-based.

The climate oriented ICON-A (ICON atmosphere) physics package, as used in Giorgetta et al. (2022), can not be used in
this study as not all components are yet optimized for performance on the NEC Aurora. However, ICON-NWP and ICON-A
use the same dynamical core, that uses roughly half of the run time, and similar data structures and programming model. Thus
we assume that the principle performance aspects of both physics packages are quite similar, so that we use the NWP package
to study hardware differences that would also apply to the ICON-A package.

In the chosen configuration, ICON runs the non-hydrostatic dynamical core, a MIRUA-type (Miura, 2007) horizontal trans-
port scheme with linear reconstruction for hydrometeors and combination of MIRUA with cubic reconstruction and a flux-form
semi-Lagrangian horizontal advection for water vapour, a piecewise parabolic method for vertical tracer transport, a prognostic
turbulent-kinetic-energy scheme for turbulent transfer (Raschendorfer, 2001), TERRA as the land surface model with tiles
(Schrodin and Heise, 2001; Schulz, 2006), a single-moment five-component microphysics scheme (Doms et al., 2011; Seifert,
2006), a shallow and deep convection scheme (Tiedtke, 1989; Bechtold et al., 2008), a sub-grid scale orographic (SSO) drag
scheme (Lott and Miller, 1997), a non-orographic gravity wave drag (GWD) scheme (Orr et al., 2010), the ecRad radiation
scheme (Hogan and Bozzo, 2018; Rieger et al., 2019), and other computationally less expensive schemes. Different time step-
ping is used for different components. The so called fast-physics time step $\Delta t$ is used for tracer transport, numerical diffusion
and physics parametrizations such as turbulence, TERRA, microphysics. The convection scheme is called with $2\Delta t$. SSO and
GWD parametrizations are called with $4\Delta t$. Radiation is called with $6\Delta t$. The less frequent calling frequencies reflect the
relatively slower changing rates of the parameterized processes. However, the dynamical core is called at usually 5 sub-steps
of the fast time step $\Delta t$ and the number of sub-steps is increased automatically to adapt for rare cases of very large orographic
waves. Such waves would otherwise be numerically unstable.

Besides a lower calling frequency, the cost of the radiation scheme is further reduced by computing the radiation on a
horizontal grid of reduced resolution. The reduced grid has twice the grid spacing of the original grid and the grid points are
redistributed over the MPI processes so that they are balanced evenly in longitudinal and latitudinal direction. This means each
process computes a similar amount of day- and night-time grid points as well as similar amount of winter and summer points.

The ICON model is set-up globally and the horizontal resolution determines $\Delta t$. $\Delta t$ is set to 6 and 3 minutes for grids with
a grid spacing of 40 km (R2B6) and 20 km (R2B7) respectively. The model is configured with 90 levels in the vertical for





all horizontal resolutions. The model is initialized with non-idealized data from the NWP data assimilation cycle. The data assimilation was run directly for the target grid so that no initial interpolation or extrapolation of the data is required.

ICON offers a two-way nesting option to study selected regions at a higher spatial resolution. The nesting uses an additional horizontal domain that has half the grid spacing of global grid and that is limited in space. The nest is informed from the global
domain at its boundaries and feeds back in its interior after doing two integrations of length $\Delta t/2$. The convection and SSO schemes are also called twice as often as on the global domain, however the stepping of GWD and ecRad remains unchanged. In the vertical, the nest is limited to the lower 60 levels of the global domain and the initial boundary conditions at the nest top are derived from the global domain as described by Zängl et al. (2022).

The reader is referred to chapter 3 of Prill et al. (2023) for further details on the ICON NWP model.

The experiments of this study are based on the code of the 2.6.6 release candidate of ICON. Specifically the f1a815e27c git commit has been used initially but that version is indistinguishable from the 2.6.6 release in terms of the reported performance. The performance has not changed in the commit 7cc6511e76 of the 2.6.7 release candidate and the 7cc6511e76 version was also used as adoptions were necessary due to updates of the HPC software stacks. The ICON binary used in Section 3 for benchmarking the GPU has been compiled with code-inlining enabled.

## 3   Multi-Node Scaling

Multi-node parallelization in ICON is based on domain decomposition. This imposes scaling limits in both strong-scaling and weak-scaling. In this section we discuss the scaling of the entire time loop on different hardware architectures. This discussion will demonstrate the influence of the application scope, such as domain size and time-to-solution constraints, on the choice of the most suitable hardware.

The scaling of ICON is assessed using an experiment series that is based on the numerical weather prediction (NWP) physics package of ICON. The NWP physics is well suited for kilometer scale simulations and has been adopted for all three hardware architectures. The scaling of this test experiment shows the same basic scaling characteristics as discussed in Giorgetta et al. (2022) for a climate simulation oriented experiment. This means that ICON can be scaled well over multiple nodes by increasing the horizontal resolution. The wall clock time per model time step is almost constant as long as the number
of grid cells per node is kept constant. Such fine *weak-scaling* is observed for CPU, GPU and vector systems. It should be noted, however, that a doubling of the horizontal grid resolution requires a doubling of the number of time steps according to the Courant–Friedrichs–Lewy condition.

*Strong-scaling* limits on the other hand, set an upper boundary on the maximum throughput that can be achieved on a particular architecture when using high numbers of nodes. ICON's strong-scaling is analyzed in the following using an R2B7
global grid (20 km global horizontal grid spacing, 1310720 cells) and a regional grid in the nested domain (10 km spacing, 212760 cells). This resolution is chosen so that MPI communication is required on all architectures as ICON requires more memory than available in a single GPU or vector engine.





For any given number of compute units, the latest NEC VE30A computes the solution the fastest (Fig. 3). However, when comparing among the older generation of hardware, and if a smaller number of nodes is sufficient to run a simulation in a given time limit (typical case for NWP ensemble predictions), then NVIDIA's A100 outperforms the VE10AE and EPYC 7763. In climate applications however, long periods are simulated. Here, the fastest time-to-solution or highest SDPD (Simulated Days Per Day) matters. Among the older generation of hardware, the VE10AE performs best with up to 256 vector engines. The CPU system can outperform the VE10AE only slightly when using 1024 CPU sockets, however such setup would be much more costly in terms of hardware and power.

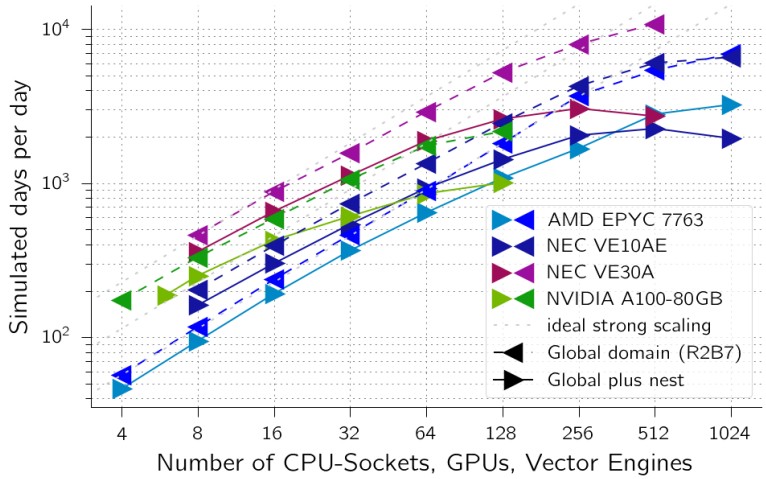

**Figure 3.** Strong-scaling. Based on runs with global domain (R2B7, 20 km grid spacing with 1310720 cells) without nesting and runs with the global and a nested domain (10 km, 212760 cells). Time step: 2 minutes, radiation (ecRad) called every 16 minutes on a globally-balanced horizontally-reduced grid.

The strong-scaling limit can be explained by the number of cells computed by each compute process. When more and more nodes are used for the same problem sizes, the number of cells per compute process decreases. In a nested setup the size of global as well as the nest domain matters as both domains are distributed equally over all compute processes. As the nest is about 8 times smaller than the global domain, the number of nest cells per compute process limits the scaling for the nested setup. Therefore, the performance degrades earlier with nesting than without nesting. For example the GPU setup without nesting does gain very little speedup when using more than 64 GPUs. With 64 GPUs there are on average 20480 prognostic cells per GPU (Fig 4). This means that each of the 3456 double processing units on an A100 handles only no more than 6 cells a horizontal loop/kernel over all cells. With even more GPUs there is less computation within each kernel that could hide memory access latency. A similar argument can be made for the VE10AE vector engines which saturate at about 512 VEs. In that case there are about 320 cells per process. This hardly fills the vector length of 256 more than once. The peak throughput of the VE30A system is at about 250VEs which relates to the same number-of-cells to vector-length ratio as the VE30A has twice as many vector units as the VE10AE.



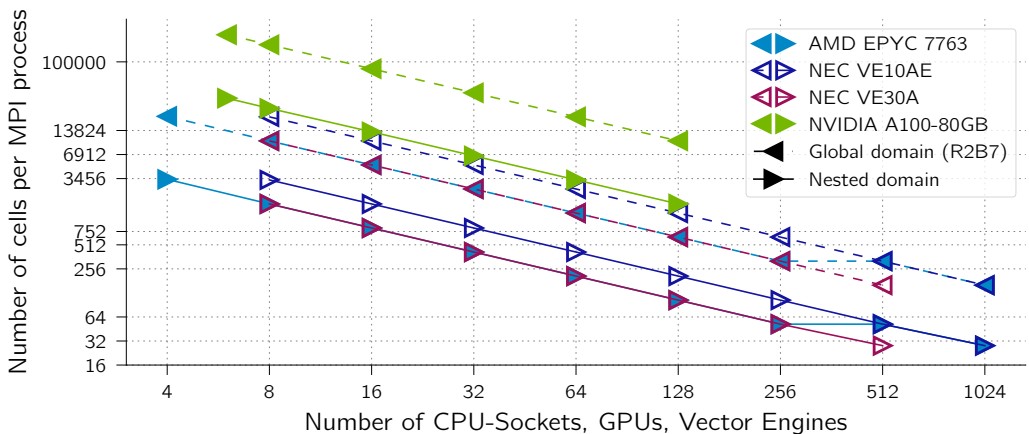

**Figure 4.** Number of cells in each domain per MPI process. 8 processes per NEC VE10AE, 16 processes per NEC VE30A, 1 process per NVIDIA A100, 16 processes per AMD EPYC 7763 socket with 8 OMP threads each (hyper threading) in configurations with less than 512 sockets, and 8 processes per CPU socket in the 512- and 1024-sockets configurations (no hyper threading).

Almost perfect scaling can be seen for all architectures when just a few nodes are used. The GPU setups scales well up to 8 GPUs with and without nesting. The vector engine setup scales well at 16 and 32 vector engines with and without nesting, respectively. The CPU setup scales well up to 32 CPU sockets with nesting, 128 CPU sockets without nesting. These results are transferable to other resolution by scaling the number of resources linearly with the number of grid cells due to the good weak-scaling of ICON (Giorgetta et al., 2022). The total running time of MPI-parallel programs mainly consists of the time for calculations and the MPI-communication overhead. Furthermore, the time for computations is affected by workload imbalance. With increasing number of nodes the overhead is increased, which means that both strong- and weak-scaling have limits (Neumann et al., 2019). Thus, the performance of a single node becomes the crucial factor in overcoming these limits.

## 4 Single-Node Performance

The investigation of the single-node performance is based on the architectures with their theoretical performance metrics as described in Section 2.1. The investigation will be divided into two parts. First, the results of the LINPACK and HPCG (High Performance Conjugate Gradients) benchmarks will be compared and the performance of characteristic numerical kernels of the climate code ICON on the different architectures will be examined subsequently. The purpose is to compare the difference between the promised theoretical performance benefit of an architecture and the actual performance gain in a real application.

The performance evaluation is based on the roofline model (Williams et al., 2009). The roofline model often serves as a visual method for evaluating the performance of high-performance computing systems. The model uses the peak floating-point performance (or arithmetic performance) and the peak memory bandwidth of the hardware as boundaries. The achieved compute performance and compute intensity of an application is set in relation with its theoretical bounds. The roofline model





helps to recognize hardware limits and to determine if an application is compute bound or memory bound. The horizontal axis represents compute intensity (Flop/Byte), the vertical axis shows performance in Flop/s. The bandwidth limit of the hardware is calculated as the product of the architectures peak memory bandwidth and the operational intensity, serving as an upper performance bound for memory bound applications. The horizontal ceiling of the roofline model is given by the theoretical maximum computing power. For the slanted bandwidth limit, we use both the theoretical limit and the value of the stream benchmark (cf. Fig. 6).

For the measurements, the executables are generated with different compilers and compiler options. Compilers and options are selected in such a way that the best possible performance is achieved on one full node for each specific hardware. Table 4 shows the possible compilers for the respective architecture, the selected compilers are marked with an asterisk.

**Table 4.** Compiler-architecture-matrix: The asterisks mark the choice with the best performance

|  | **ifort** | **gcc** | **crayftn** | **nec** | **pgfortran** | **nvfortran** |
|---|---|---|---|---|---|---|
| CPU | ✓ * | ✓ | ✓ |  |  | ✓ |
| GPU |  |  |  |  | ✓ | ✓ * |
| NEC |  |  |  | ✓ * |  |  |

## 4.1 HPL and HPCG

The configurations for the HPL (High Performance LINPACK) and HPCG tests performed on the various architectures are shown in Table 5.

**Table 5.** Settings of HPL and HPCG tests. T/V: Wall time/encoded variant; N: order of the coefficient matrix A; NB: partitioning blocking factor; P: number of process rows; Q: number of process columns.

| HPL | T/V | N | NB | P | Q | HPCG | Domain | Process grid | Duration [s] |
|---|---|---|---|---|---|---|---|---|---|
| CPU | WR00L2L2 | 114688 | 128 | 8 | 16 | CPU | 128x128x128 | 8x4x4 | 1800 |
| GPU | WR00L2L2 | 131072 | 288 | 2 | 2 | GPU | 256x256x256 | 2x2x1 | 1800 |
| VE10AE | WR13R4R16 | 207132 | 246 | 2 | 8 | VE10AE | 384x576x1504 | 4x4x4 | 1800 |
| VE30A | WR13R4R16 | 292986 | 246 | 2 | 16 | VE30A | 768x576x1504 | 8x4x4 | 1800 |

The NVIDIA HPC-benchmark 21.4 containers were used to perform the HPL and HPCG benchmarks on a GPU node. The scripts `hpl.sh` and `hpcg.sh` had to be adjusted within the containers to make both run efficiently on a full GPU node. The results are summarized in Table 6.





**Table 6.** Results of HPL and HPCG benchmarks in TFLOP/s

| Benchmark | CPU [TFLOP/s] | GPU [TFLOP/s] | VE10AE [TFLOP/s] | VE30A [TFLOP/s] |
|-----------|---------------|---------------|-------------------|-----------------|
| HPL       | 3.08          | 37.98         | 17.78             | 35.32           |
| HPCG      | 0.04          | 1.13          | 0.98              | 2.09            |

A huge performance difference between HPL and HPCG is obvious (cf. Table 6). The performance loss on the CPU is a factor of 77, on the GPU it is a factor of 33.61 and on the NEC VE10AE it is a factor of 18.1 (VE30A: 16.9). These measurements and the observed efficiencies are in line with a comparison based on single devices (Takahashi et al., 2023). The performance difference between HPL and HPCG shows the impact of irregular memory access patterns which are used in the HPCG benchmark. It should also be noted, that the HPL benchmark simply aims to measure the maximum floating-

point execution rate of the architecture by solving a dense system of linear equations, whereas the HPCG benchmark uses sparse matrix-vector multiplication. The memory access patterns of many real world applications like the climate and weather prediction model ICON come closer to that of the HPCG benchmark. This will be further affirmed in the measurements in Section 4.2.2.

### 4.2    Principle Analysis of ICON kernels on different HPC architectures

Section 3 assesses the strong-scaling of ICON on multiple nodes. The experiments revealed that GPUs potentially perform better and result in lower run-times, compared to hardware of the same age, as long as the parallel capabilities of the GPUs can be fully exploited, i. e. as long as there are enough grid cells per MPI process. In this section, we want to answer the question of how much the architectures exploit their potential in the single-node case and which performance characteristics such as bandwidth or compute intensity are responsible for the performance. For this we use the roofline model which is a fairly simple

but often very suitable performance model on HPC systems and associated software.

#### 4.2.1    Measurements

**CPU**

The measurements are performed with LIKWID (Treibig et al., 2010), although it was only used to read the corresponding hardware counters. The ICON timers were used for the runtime measurements and the metrics were calculated accordingly.

For the flops, the `RETIRED_SSE_AVX_FLOPS_ALL` event of the PMC counter was measured and the flops were calculated as described in the performance group `FLOPS_DP`. For the bandwith, the `DRAM_CHANNEL_0:7` event of the `DFC` counter was summarized and the bandwith calculated as described in the groups `MEM1` or `MEM2`. Note, that the memory measurements of the DRAM channels 1–7 required two different runs, as there are only four channels to measure the counters on. For the compute intensity the quotient of flops and memory bandwidth was used. The ICON binary was built using the Intel®Fortran

Compiler 2021.5.0 20211109 with the performance optimization flag `-O3`. The application runs with hybrid MPI and OpenMP.





A small parameter study showed the best performance for 32 MPI ranks with 4 OpenMP processes each and a `nproma` value of 8. The value of the run-time tuning parameter *nproma* represents a blocking length for array dimensioning and ideally achieves better memory access (cache blocking). It is therefore dependent of the architecture and a typical value for the AMD CPU used in this study is 8, for vector processors it depends on the length of the vector registers and is much larger. Multithreading was

disabled for better performance and the tasks were distributed across the cores using SLURM via plane distribution. The plane size was also 8. The stream benchmark for the roofline ceiling resulted in a value of about 340 GB/s for the Triad benchmark.

**GPU**

To evaluate the performance of the A100 GPUs two NVIDIA tools are used, Nsight Systems and Nsight Compute. Nsight Systems reports which kernels are launched in which ICON timer region (see also fig. 5). Typically multiple kernels are

launched within even small timer regions but the actual kernel computations doesn't necessarily end (or even start) in a given timer region as most kernels are launched asynchronously. The assignment of kernels to timers, their ratio compared to the overall kernel computation of kernels launched in that interval, and the overall timer duration itself are obtained from the Nsight System profile.

The performance and computational intensity of each individual kernel is obtained by a second run with Nsight Compute.

Since the overhead to a normal experiment run is quite substantial and the current setup requires a single GPU run at the moment, only the first invocation of each kernel is investigated. It is also assumed that the computation paths and amount of data for all other invocations are comparable.

For each investigated timer, the kernel performance is related to the duration from the first kernel start to the last kernel end of all kernels launched in that timer region. The computed GPU compute intensity does not include the time loss due to the

first kernel launch overhead. Using $B_1$ from Figure 5 as an example, this is the difference between the start of $K3_1$ and the start of $B_1$.





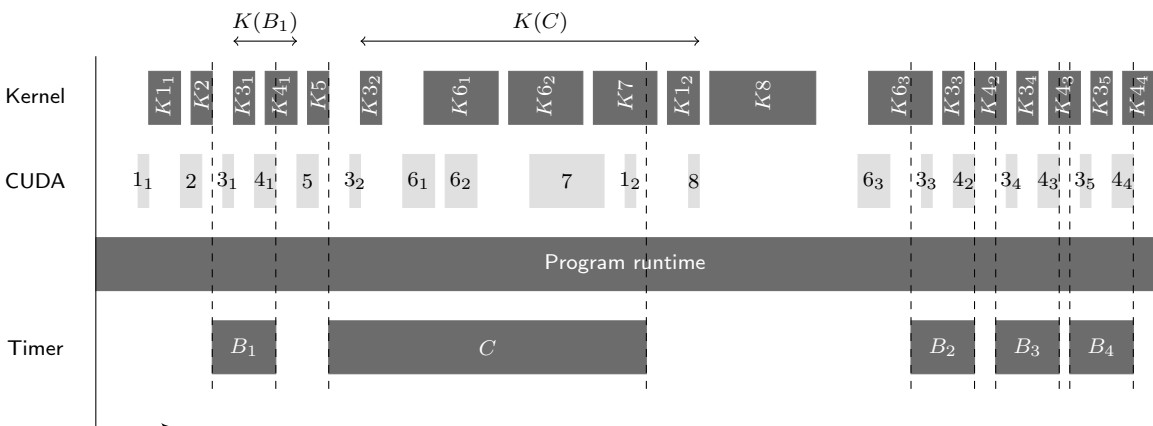

**Figure 5.** Exemplary output of Nsight showing Kernel activity in the upper row, CUDA kernel launches in the second row and custom timer regions in the bottom row. To evaluate the GPU performance in one region like $B_1$, the respective kernel performances within that region are obtained by all kernel performances of kernels launches in that region (e.g. kernels $K3_1$ and $K4_1$).

**Vector Engine**

For the NEC SX-Aurora TSUBASA ICON has been used in a "hybrid MPI" mode with initialization and I/O processes running on the x86 vector host CPU while the computational processes were launched on the vector engines (VE). The VE executable

has been built using NEC MPI 3.5.0 and NEC compilers 5.1.0; OpenMP has not been enabled.

Performance data for the experiment `ICON_09_R2B6N7_oper_EPS_noIAU` have been collected from jobs using eight VEs (8 cores each) with an `nproma` value of 752. NEC's performance analysis tool `ftrace` can provide data like computational performance or Byte/FLOP ratios for a whole program or specific code regions.

It should be noted that the number of floating point operations reported by ftrace differs from the numbers reported by

Likwid or Nsight if conditional code is involved: for a loop containing an `IF/ELSE` construct the vector processor executes both branches for all iterations and uses a logical mask to assign only the necessary results to the respective variable, i. e. more floating point operations than necessary are executed, and this number is used by `ftrace` to calculate e. g. the performance in Flop/s. As the time used in this calculation is the sum of the time to calculate the necessary results and the time to calculate the unused results we assume that the Flop/s numbers shown by `ftrace` are also representative for the necessary part only

and therefore can be used in comparison to results from other tools which only use the number of necessary floating point operations.

#### 4.2.2 Single-Node comparison of the architectures

In this section we will analyze typical ICON components using the roofline model. The ICON kernels under consideration are the non-hydrostatic solver (`nh_solve`), the radiation (`nwp_radiation`) and the transport schemes (`transport`).

Figure 6a shows the achieved computational performance (in GFlop/s) together with the arithmetic intensity (in Flop/Byte)



for these kernels and the same data for the HPCG benchmark (Dongarra et al., 2016) on the different processor types. With a considerably low arithmetic intensity, all of the data points are situated in the area below the *bandwidth ceiling* of all processor types (the solid diagonal lines show the theoretical maximum memory bandwidth, the dashed lines show the bandwidth of the Triad operation of the STREAM benchmark (McCalpin, 1995) for each architecture). This means that the possible maximum

performance of the kernels is limited by the memory accesses inherent to the used algorithms, not by the theoretical peak computational performance of the processor. The very similar performance of the HPCG benchmark (which is intended to test the effect of memory limitations on computational performance) further corroborates this. It should be noted that the HPCG benchmarks performance is based on the value calculated in the benchmark and not the measured value as described above for the ICON kernels. The achieved computational performance is close to the respective bandwidth ceiling for all

architectures which means that there is little potential for further optimization in these code parts (cf. 2.2.1 for a discussion of code adaptations for the different architectures).

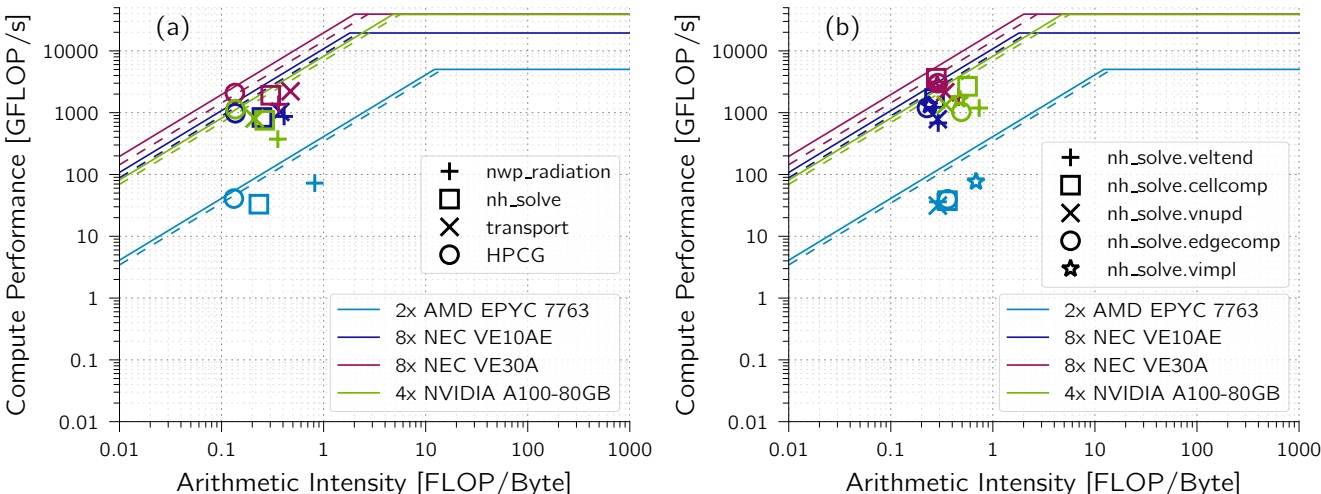

**Figure 6.** Roofline Model for selected ICON kernels and HPCG benchmark (left) and the ICON kernels below `solve_nh` (right) in a single-node comparison. The dashed lines represent the memory bandwidth (Triad) calculated with STREAM benchmark.

Yet it is to be noted that none of the analyzed kernels saturates the theoretical memory bandwidth limit given their arithmetic intensity, and most of the kernels don't even quite reach the benchmarked memory bandwith bound. This can be explained partly by the fact that the memory access patterns of the respective algorithms are not ideal. Variables in ICON are stored in

contiguous memory for each physical quantity. Most operators use multiple variables which leads to jumps in memory access. Cache misses are even more frequent in operators that operate on horizontal neighbours. As ICON uses an unstructured icosahedral grid, horizontal neighbour relations cannot be exploited in the memory layout. Despite that, the achievable performance would still benefit from a higher peak memory bandwith of the architecture of course. Another reason is that due to the decomposition of the simulation domain, each of these large code regions contain a certain amount of MPI communication,

which reduces the arithmetic intensity. To separate its influence on the performance data shown in Figure 6a from the compu-





tational performance, Figures 6b and 7b show the performance and arithmetic intensity for sub-regions of the `nh_solve` and `transport` kernels which do not contain as much communication, so with this communication time partially excluded, both values are higher for the `nh_solve` sub-kernels, but their computational intensity still characterizes them as memory-bound (Fig. 6b).

This can also be seen in Figure 7b where the vertical advection flux calculation (`adv_vflux`) demonstrates better performance than the horizontal one (`adv_hflx`) as the horizontal advection code still includes some MPI communication. We observe that the position of (`adv_vflux`) is further to the right and therefore higher on the plot. This analysis underscores the importance of fast memory access in enhancing the performance of these kernels. The `transport` kernel comprises both `adv_vflux` and `adv_hflux` and is also affected by the MPI communication.

The roofline plot shown in Figure 7a reveals that the `nwp_radiation` (ecRad) kernel exhibits very low performance, being still far away from the bandwidth ceiling. This is possibly due to factors like different loop ordering or vector lengths. The CPU performs relatively well for its limits compared to the strong under-performance of the GPU and the VE.

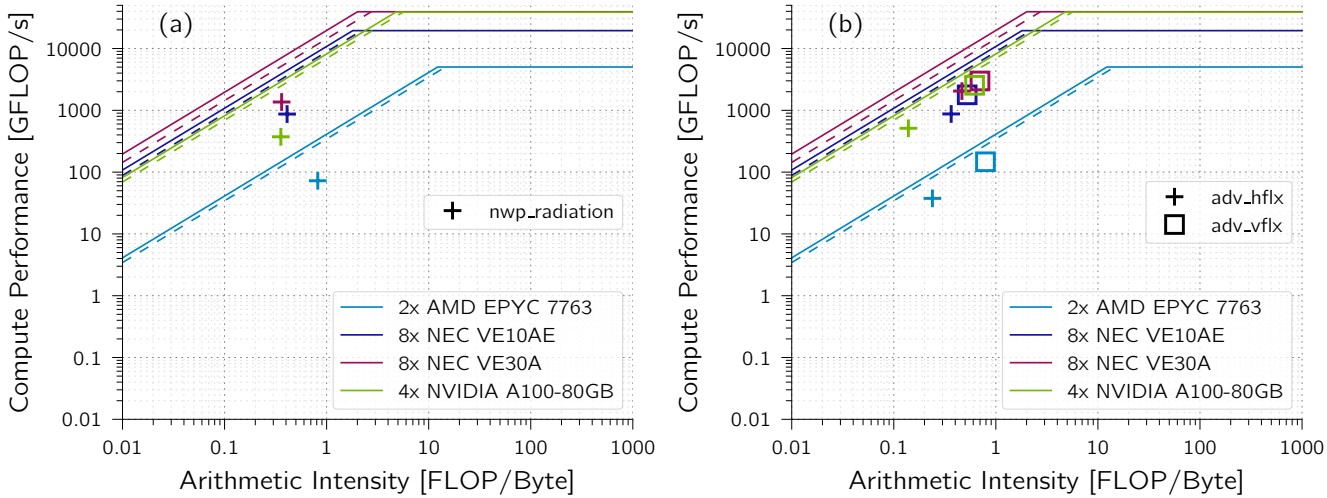

**Figure 7.** Roofline Model for the ICON kernel `nwp_radiation` (left) and the kernels `adv_hflx` and `adv_vlfx` of the `transport` scheme (right) in a single-node comparison. The dashed lines represent the memory bandwidth (Triad) calculated with STREAM benchmark.

    In all rooflines we see that GPUs and VE30A have the highest peak performance ceiling. However this cannot be utilized because of the low computational intensity of the ICON kernels. The hardware with the highest memory bandwidth is the

NEC, as seen by the corresponding ceiling in the roofline, and all kernels on this architecture show the best actual compute performance. The same holds for the different parts inside the kernels. This is highlighting that for ICON, maximum bandwidth limit is more crucial than computational peak performance.

    To fully exploit the A100 memory bandwidth, the number of parallel computations, basically the number of grid points, has to be multiple of the number of available processing units. The streaming multiprocessor can hide memory access latency by





swiftly switching the stream from one that is waiting for input to another that is ready for computation, this effectively was also evident in Section 3.

Overall this means that the ICON software offers too few calculations compared to the necessary data transfers so that the computational power of the processing units cannot be exploited in an optimal way. The speedup over the CPU observed in GPUs for many of the investigated kernels is therefore mostly to be attributed more to High Bandwidth Memory (HBM)

than computational peak performance. Due to this the actual speedup for the ICON kernels falls short of what would have been expected from the theoretical performance values or the results of the HPL benchmarks (cf. Table 6). It is important to recognize that we can also achieve better performance by raising the achieved compute intensity, which is primarily software-driven and varies with code porting to a specific hardware.

## 5 Outlook

Not only the ICON community is currently on the way to using current and upcoming Exascale systems for high-resolution simulations. The simple cloud resolving E3SM atmosphere model for example achieved a performance of 1.26 simulated years per day (SYPD) running a setup with a horizontal resolution of 3.25km and 128 vertical levels on the entire Frontier system (No. 1 in the current Top500 list), see Taylor et al. (2023). Frontier is the only Exaflop system in the Top500 list. It delivers a theoretical peak performance (Rpeak) of 1.7 EFlop/s and the maximal LINPACK performance (Rmax) is at 1.19 EFlop/s.

However the power consumption is at 22.7 MW.

For ICON within Destination Earth, the situation is similar. Running a coupled atmosphere-ocean setup with a horizontal resolution of 5 km and 90 vertical levels on 158 GPU nodes of the LUMI system results in a throughput of about 100 simulated days per day. LUMI is the first European system in the pre-exascale era and delivers a theoretical peak performance of 0.53 EFlop/s and a maximum LINPACK performance of 0.38 EFlop/s at a power consumption of 7.1 MW. The above figures show

that with the current setup 1 SYPD at 5 km resolution is still achievable on a fraction of LUMI (estimated 580 nodes), while for the 2.5 km setup further optimisations may be needed, as halving the horizontal resolution results in an 8-fold increase in resources.

Based on our principal analysis of the ICON kernels and the fact that ICON is memory bound one can estimate that a global coupled atmosphere-ocean simulation using ICON at 1 kilometer resolution and 1 SYPD performance would certainly also

require almost the full scale of future Exaflop supercomputers. This means that energy efficiency becomes a crucial aspect, both in terms of making the costs of such simulations affordable and in terms of the carbon footprint.



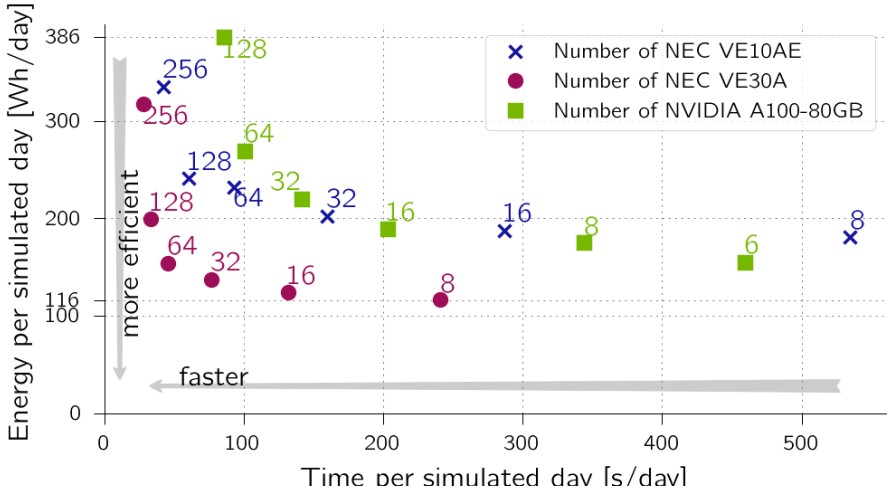

**Figure 8.** Throughput and Energy used for time loop. Global R2B7 plus Nest. Electrical power reported by `nvidia-smi` (NVIDIA) and `veda-smi` and `vecmd` (NEC), both excluding the host CPU.

While it is undisputed that GPU-based systems are very well suited for dense computing and, in particular, machine learning applications, the tide is turning for sparse computing. This can be seen to some extent in the HPCG list, where systems such as Fugaku or SX-Aurora TSUBASA-based machines occupy top positions - even if the associated energy consumption is unfortunately not specified. Figure 8 shows a comparison of the power efficiency of ICON on two such architectures, a system equipped with NVIDIA A100 GPUs and a NEC Aurora system. The X-axis shows the time required for a simulated day in seconds, while the Y-axis shows the energy required in Watt-hours(Wh) per simulated day.

When the number of processing units (GPUs or Vector Engines) used is between 6 and 16, which is the case in low resolution simulations, then the energy to performance ratio is better on GPUs (see the right part of Figure 8). This observation changes by almost a factor of 2 when using the latest VE30A system, but an adequate comparison should be drawn to latest Nvidia Hopper system then. However, when the number of computing units exceeds 64, it is on the one hand remarkable how exponentially the energy consumption increases. On the other hand, it shows that an architecture that is suitable for the problem (in this case the NEC Aurora system) outperforms the prevailing opinion on GPUs in terms of energy efficiency.

High-resolution climate simulations at the 1km scale will in any case require significant high-performance computing resources. Energy efficiency will be a key concern to ensure that resource utilisation does not lead to exorbitant energy consumption. This means that future work should look not only at runtime but also at energy metrics such as energy to solution. The ICON model, with its various computational kernels, is primarily memory-bound as shown, but the performance benefits of graphics processing units (GPUs) and vector processing systems vary significantly between different kernels. And in some cases (when e.g. memory access is highly scattered) even a CPU-based implementation (possibly relying on HBM) might outperform the rest. For this reason, ARM architectures (such as the A64FX or Neoverse) will increasingly have to be analyzed for their suitability for ICON in the future. This variance underscores the importance of code refactoring, a process that is





critical to optimising resource allocation. By disentangling code, it is possible to achieve a more flexible and efficient use of HPC resources, balancing the need for computational power with the need for energy efficiency.

## 6 Conclusions

From section 2 we conclude that the monolithic software design of ICON is poorly suited to the variety of heterogeneous architectures that exist in modern High Performance Computing systems. Even the portability of individual code segments to all possible architectures has been practically difficult to achieve. This limitation underlines the importance of code modularisation, which is being addressed, for example, in the WarmWorld project.

In section 3, it becomes evident that the scalability of ICON on various architectures encounters a limit relatively early.
Merely adding more processing units (PUs) as a "brute force" approach fails to significantly enhance speedup, leading to a point where there are too few cells per PU. This necessitates extracting more from single-node performance, as simply increasing the number of nodes also constitutes an energy waste if the performance of each single node is not optimally utilized. From the multi-node scaling analysis we can conclude furthermore that VE30 computes the solution the fastest. The NVIDIA A100 is faster than VE10, as long as there are enough grid cells per MPI task. The GPU and both of the Vector
Engine architectures show an expected speedup compared to the CPUs. However, not all components of ICON are yet ported to OpenACC or optimized for vector architectures.

The single-node investigations in section 4 show a clear drop in maximum performance from the theoretical manufacturer specifications, to the HPL benchmark, the HPCG benchmark and the ICON kernels respectively. The measurements and the comparison with the performance achieved in the HPCG benchmark indicate that this benchmark is a far more representative
way of assessing the achievable performance for ICON on an HPC system than, for example, the HPL. This means, that looking on the exascale systems on the Top500 list doesn't show any suitable system for exascale performance with ICON yet. Since the computational peak performance limit is still far away given the compute intensity in the single-node measurements, it is reasonable to assume that ICON benefits more from architectures that enable more throughput and less from architectures that benefit from a strong computational peak performance alone.

Finally, the energy efficiency of individual HPC systems for the ICON kernels under consideration also supports this observation. Since the main requirement is not computational power but memory bandwidth, GPU systems are often not necessarily the most energy efficient solution for the ICON model, contrary to the usual trends in the Green500.

*Code and data availability.* Simulations were done with the ICON 2.6.6 branch as described in section 2.2.2, which was available to individuals under restricted licenses. This code version is close to the current public version of ICON, which is available under BSD-3C license
(ICON partnership (DWD; MPI-M; DKRZ; KIT; C2SM) (2024). ICON release 2024.01. World Data Center for Climate (WDCC) at DKRZ. https://doi.org/10.35089/WDCC/IconRelease01) and also contains the kernels used within this paper. The setup of the test experiments is at Jacob (2024).



*Author contributions.* Panagiotis Adamidis and Erik Pfister structured and led the study, with Panagiotis Adamidis being responsible mainly for Section 2 and Erik Pfister for Section 4. Erik Pfister also carried out the CPU measurements for the ICON components whereas Panagiotis

Adamidis provided the main portion of contents in chaper 2. Hendryk Bockelmann is mainly responsible for the content of Section 5. Dominik Zobel analyzed the GPU single node performance in Section 4. Jens-Olaf Beismann analyzed the Vector Engine single node performance in Section 4. Marek Jacob analyzed the multi node scaling in Section 3 provided the ICON test experiment configuration, and contributed to the discussion of the GPU performance. All authors contributed to the writing of the manuscript.

*Competing interests.* The authors have no competing interests.

*Acknowledgements.* This work used resources of the Deutsches Klimarechenzentrum (DKRZ) granted by its Scientific Steering Committee (WLA) under project ID ka1352.

*Financial support.* This work is partially funded by the Deutsche Forschungsgemeinschaft (DFG, German Research Foundation) under Germany's Excellence Strategy – EXC 2037 'CLICCS - Climate, Climatic Change, and Society' – Project Number: 390683824, the Bundesministerium für Bildung und Forschung (project EECliPs under grant no. 16ME0599K), and by the Deutscher Wetterdienst (DWD) under

the "Innovation in der angewandten Forschung und Entwicklung" (IAFE VH 4.1) initiative.





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
