# Peer review of "The Real Challenges for Climate and Weather Modelling on its Way to Sustained Exascale Performance: A Case Study using ICON (v2.6.6)"

_Geoscientific Model Development, 2024_

## Author Comment (AC1)

**Author's Point-by-point response to the reviewer**

1. Figure 1 is a nice-looking figure, but I don't believe it adds anything to the discussion and is in fact misleading as it isn't really saying anything of substance at all. It doesn't reflect the point made in the text: "the appropriate programming model becomes a difficult task as not all of them support all types of accelerators" (lines 67-68), but the diagram just shows a wall with each brick representing a different programming model and arrows going indiscriminately to different hardware. It would instead be useful to show which programming models work on which hardware, either through a different diagram or a table.

*We agree that the added value is not high enough and therefore removed it.*

2. Figure 2 likewise looks very nice but doesn't really contain any substance about how the ICON code is currently and how ICON-C will change the code structure and apply the different programming models. I would rather for the authors show code snippets or pseudocode to illustrate how the modules could be ported to the new architectures using the different programming models, and how would this be done, e.g. manually or using a code writer? Other models, such as the LFRic weather and climate model (https://www.metoffice.gov.uk/research/approach/modelling-systems/lfric), are using tools such as PSyclone (https://psyclone.readthedocs.io) to try to achieve performance portability of the Fortran source code on different architectures.

*We changed both illustrations to respect the given feedback. The switching in the current monolithic design is handled by various preprocessor macros for the respective architectures as illustrated in the updated left hand side of the figure. In contrast, the new ICON-C design now emphasizes that only one module of each kind is needed.*

3. Figure 3 is quite confusing. The ideal strong scaling curves are very difficult to see as they are a light grey colour (the same as the gridlines behind). The point type used (left- and right-facing triangles) are difficult to differentiate and the fact that the colours are slightly different for the global and nested domains, but not different enough, makes it hard to unpick this. The different line styles are also not discussed. The authors could consider making the plot much larger, using very different point styles, mentioning the different line styles, and perhaps separating out the global and nested results into different sub-figures.

*We revised the plotting style of this figure. Every line has now its distinct, unique combination of style and markers. The line style encodes the processor type while the markers denote the domain. Both are explained in the legend. Colors are used to emphasize the different line styles such that they identify the different processors in the same way as in the roof line plots later on.*

4. Figure 4 is very interesting given the results in Figure 3 and the core numbers in Table 2. I would suggest highlighting the number of cores in the discussion around GPUs for Figure 3. It might also be helpful to highlight on Figure 4 the number of cores for each hardware type configuration, so it can be seen when the hardware becomes underutilised.

*We modified the figure caption motivated by the reviewers comment and adjusted the figure style to match the previous figure. The new caption provides more explicit information on the compute capability of each respective processor for a single MPI process. The number of cores are discussed in line 186–205.*

5. For Figure 5 I'm a bit unsure if the timing for B1 as given in the example will include to the end of the K41 kernel, i.e. outside the end of B1.

*To clarify this, additional vertical lines were added in the top area. It should now be clear, that $K(B_1)$ extends from the beginning of $K3_1$ to the end of $K4_1$ and $K(C)$ from the beginning of $K3_2$ to the end of of $K1_2$.*

6. The plots in Figure 6 are very hard to see as the points are large and the lines are close together and overlapping. The points are also all clustered around a small section of the graph, but the scale is much larger in X and especially Y, mainly to include the legend. I would suggest plotting each point on its own graph, making a large multi-figure plot, and zooming in as much as possible onto (0.1:100, 1:50,000) ranges to allow the relevant areas to be seen as clearly as possible. I would likewise do the same for Figure 7 by combining Figures 6 and 7 in a single multi-panel plot.

*We agree that the previous versions made it hard to see the differences. But since we also wanted to express where the results are compared to the shape of the rooflines, we decided to add a zoom-in of the interesting region for each plot.*

7. Figure 8 is also really interesting. Do you know the energy usage from the CPU runs?

*We have no energy metrics for the CPU runs as we had no conceptual approach to measuring the energy consumption of a simulation on the CPU nodes at the time of the studies.*

8. Line 17: I don't believe "System" should be capitalised here.

*Corrected as suggested*

9. Line 26: The term GPU is used without being defined (this occurs on line 40). Also, the discussion here is around x86 hardware being combined with GPUs, Vector, or ARM systems. Note that superchip hardware, such as NVIDIA's Grace-Hopper (https://resources.nvidia.com/en-us-grace-cpu/grace-hopper-superchip) is an ARM-GPU system, i.e. there is no x86 CPU, where the CPU itself is instead ARM-based.

*The definition of GPU was put after the first use of the acronym as suggested. Regarding ARM-based systems, we are referring to the A64FX processor of the Fugaku supercomputer, where the CPU is ARM-based.*

10. Line 345-6: I think this sentence needs to be rephrased: "Not only the ICON community is currently on the way to using current and upcoming Exascale systems for high-resolution simulations." - do you mean something like "ICON is not the only model that is currently on the way to using current and upcoming Exascale systems for high-resolution simulations."

*Corrected as suggested*

11. Line 356-7: Perhaps I'm missing something, but wouldn't halving the horizontal resolution result in a 4-fold increase in necessary resources, rather than 8-fold? I'm assuming that the level structure remains unchanged.

*Halving the horizontal resolution results in an 4-fold increase in grid points and requires doubling of the required time steps as noted in Section 3 (lines 170–173 of the manuscript). Thus the required resources increase is 8-fold.*

**Additional editorial changes**

- As requested during last submission the reference list is now compiled according to the GMD standard.

---

## Author Comment (AC2)

**Author's Point-by-point response to the reviewer**

1. The paper also attempts to justify the need to modularise the code, but this effort is already underway in the WarmWorld project, and it is unclear how the evidence in terms of performance results presented in this paper directly motivates this, given that the model development and optimisation on the difference architectures preceded this study, in particular by the reference Giorgetta et al., 2022.

*We acknowledge the modularization efforts of the WarmWorld project in the first paragraph of our conclusion. Our aim in Section 2.2.1 is the performance portability of the current monolithic code base of ICON. Modularization is only one aspect of it as well as being able to use different programming frameworks (like Kokkos, HIP, ...). Also we adjusted our first figures to better represent our ideas.*

2. Further, framing the performance comparisons of ICON to HPCG and HPL in Table 6 and related text, is also not new, and replicated from the study previously mentioned.

*Assuming the mentioned study is also Giorgetta et al., 2022, the $R_{\max}$ values (from LIN-PACK/HPL, no HPCG results were available then) for the whole machines are used to normalize the measured values for inter-machine comparisons. In contrast, we conducted HPL and HPCG measurements on a single node basis to compare the different architectures.*

3. The start of the paper presents the work as being rather time limited. For example, including the version of the software in the title implies, to me, that the work is only relevant to this specific version, and that is it may not be relevant for future versions, which I do not believe is the intended message.

*As suggested by the editor we added the version number of the model in the title to comply with the requirements for a Developement and Technical Paper at GMD (see e.g. `https://www. geoscientific-model-development.net/about/manuscript_types.html#item2`). Specifiying a version number also allows interested readers to reproduce the results we presented. Nevertheless, we think that our work also has relevance for some future versions. For example, as mentioned in our code and data availability statement, the source code of this ICON version (2.6.6) – with which the simulations were done – is close to the current public version of ICON, which also contains all the kernels investigated in this paper.*

4. Line 9 states that domain-decomposition parallelism reaches scaling limits, however, given the rest of the paper, this should be revised to more clearly state the distributed memory decomposition of the spatial domain.

*We clarified this in the updated version of the manuscript.*

5. On line 9 and the following, the statements that domain decomposition reaches scaling limits implies the need for single node performance is unclear.

*Exascale computers are massively parallel machines consisting of several thousands of nodes. In order to exploit their performance, the application has to scale well. Nevertheless, scaling (based on domain decomposition) has its limits which means at some point using more nodes gives no benefits anymore. Improving single node performance can be more beneficial since there are still improvement potentials.*

6. On line 41, the authors state that GPUs have dominated the Top500 list since 2015. The authors should clarify this to refer to share of compute performance, or, in the highest ranked systems, as even in the most recent Top500 list, there are more individual systems without GPUs than with and they therefore do not dominate in terms of number of systems. In short, be specific in how GPUs are dominating the list.

*Indeed the presence in the top of the Top500 list was meant and we clarified this in the updated version of the manuscript.*

7. On line 42, it would be good to include examples of current vector processors in use other than the NEC SX-Aurora. It has probably been 20+ years since the last true vector machines (e.g., Cray) . If the authors are implying use of SIMD vectorization in CPUs, then this should be made more specific.

*We are not referring to SIMD vectorization but the (pipelined) vector processing which is to our knowledge only done by the NEC SX-Aurora architecture. NEC SX-Aurora modules are used at DWD for production runs with ICON. Also, we are not referring to what the reviewer titled true vector machines but systems equipped with vector engine modules.*

8. On line 48–49, the authors justify that the NEC Aurora cards are relevant because they offer high memory bandwidth. But GPUs, and now CPUs (including the A64FX previously mentioned), also offer this memory technology. Indeed, in Table 1, high bandwidth is listed for both GPUs and the NEC cards (twice in fact for NEC!). There is some debate to be had around the "Specialities" in this table, for example OpenMP (with or without target) can run well on CPUs and GPUs as well as the NEC, along with several other parallel programming models, so the benefit the authors imply over GPUs for programming languages is spurious today.

*We removed the duplication of the NEC memory bandwidth in Table 1. Also, we are not comparing the different architectures in general but only the characteristics of the architectures we used to see, which might benefit ICON the most. In this Section (2.1) we refer to the hardware characteristics and is not about parallel programming models (they are discussed in Section 2.2).*

9. On line 46 the authors do not mention that GPUs have been used effectively for scientific computing. The way the argument is presented in this section dismisses this area of work, which again is not an authentic argument. Please rephrase this to be specific and accurate about the pros and cons of the processor categories outlined.

*We already refer to highly parallel workloads and use machine learning only as an example. Other scientific codes like solvers in the ESM community often use tightly coupled PDEs where it has to be investigated if and when parallelization is feasible. However, we added scientific computing as an additional use case.*

10. On line 65, the authors claim that vendor-specific models always give the best performance, citing an NVIDIA technical marketing document. Importantly, this document does not indeed claim that the vendor-specific model is better than other programming models, and in fact does not mention any programming model at all. As a reviewer, I also do not agree with the claim in general, as there have been several studies exploring programming models which show that the same performance is attainable in many programming models on a given hardware platform.

*Contrary to the reviewer's comment, we don't claim that the vendor-specific model always gives the best performance. We only state that the vendor-specific implementation was very efficient but tailored to the specific architecture. Also it is not portable anymore which is the main point we make here.*

11. Figure 1 is used to show that not all models support all target architectures, but in my view, the figure implies that all models support all architectures. Please consider redrawing this figure to illustrate the point. I'd encourage due diligence though, as several of the models do support all the architectures listed.

*This figure was removed because it did not add enough value to the manuscript.*

12. Line 70 is also unsubstantiated, to the paper's own detriment this time. I do no believe the claim that higher levels of abstraction imply higher performance portability and the authors do not justify this claim. In reality, there are a number of studies that show OpenMP gives the highest level of performance portability. The authors should cite these here to better argue the directive-based approach is a valid choice when striving for performance portability.

*The claim is also supported by the linked citation of Trott et al., 2022.*

13. On line 87, the authors give a 2022 reference to justify OpenACC was the only choice for Fortran GPU acceleration. However, OpenMP target was available in 2013, 10 years prior to the 2022 study. Please provide a contemporariness reference for this statement, or revise the statement.

*Offloading parts of the ICON model code with OpenACC and OpenMP was investigated internally a few years ago and at that time GPU offloading with OpenMP was not practical with any of the major compilers. We do not claim that OpenACC offloading is better than OpenMP offloading. Especially in the last years with OpenMP 4.5 and OpenMP 5.x major offloading capabilities were added which were not available to us back then. The decision to use OpenACC offloading with the ICON model had long-term implications and cannot easily be changed.*

14. On line 108, the sentence ordering humorously implies that ICON requires the code be bloated and difficult to adapt! I suggest the authors switch the order of the final two sentences in this paragraph.

*We agree and changed the order of both sentences.*

15. Line 161, and previously in the abstract, the authors mention the MPI parallelism in based on domain decomposition. It is likely obvious, but worth stating given that ICON has parallelism in other dimensions, that the domain here is longitude and latitude, and not some other dimension.

*We added this clarification in the updated version of the manuscript.*

16. The colouring of the AMD and NEC VE10AE lines in Figure 3 make them almost indistinguishable - I'd encourage the authors to explore using easily differentiable colours that are also appropriate for readers with colour vision deficiency.

*We changed the figure to allow unique identification of all plotted data.*

17. Table 2 shows that one A100 has comparable memory bandwidth to one NEC card. However, the results in Figure 3 show that the GPU performance is below that of the NEC. The text around lines 190 explain that both are starved for work in the strong scaling regime, but this question is about the results where they are not, e.g., 8 GPUs. Later, in Section 4, the authors show that the performance per GPU is very comparable in terms of the memory bandwidth bound code HPCG, and the later Roofline plots of ICON show this code also falls in this regime. Please explain the performance discrepancy. As I understand this presented work, there are no contributions to the model in terms of development, and so instead presenting a sufficient summary of the existing work on GPU and NEC optimisation becomes necessary to explain the performance the authors have measured and the Roofline analysis.

*As shown in Table 2, the difference in memory bandwidth between NVIDIA A100 and NEC VE10AE is about 2644 GB/s (roughly 25%). The performance discrepency is due to this difference which can be seen in both, Table 2 and Figure 3. As stated by the reviewer there is no other contribution of the model in terms of developement and discussing the performance is part of the added value of this manuscript (like the rooflines for specific ICON kernels in Section 4.2.2).*

18. On line 204, the authors state that the strong/weak scaling limits are overcome by improving single node performance, but I don't think this is true, as reducing the time of the computation will not resolve the scaling issues, thanks to Amdahl's Law. I think the sentiment the authors want is to improve the overall performance of the code at all scales, one needs to focus on single node performance.

*We changed the formulation to clarify our intention about overall performance optimization instead of overcoming scaling limits.*

19. On line 255, please justify why a ifort 2021 was used over a current release of this compiler. Additionally, the authors use AMD CPUs, so they should also consider use of the AMD, NVHPC, or GNU compilers, and using performance measurements to highlight the authors' choice of the Intel compilers.

*Internal measurments showed that `ifort` performs better than the tested NVHPC and GCC as was already described in Table 4. The measurements were done in 2022 with the current software stack available at that time on Levante.*

20. On line 290, the authors discuss how the NEC vector units have to execute both paths of the IF/ELSE branch, and so inflate the measure of floating point operations. The authors should present a discussion here on how the measurements of the operations for the same problem using the three different methods (LIKWID, NSight, and ftrace) are reconciled. Figure 6 shows they are similar in some cases, but not others, e.g., 'nwp_radiation' on AMD EPYC does a lot more operations than the other platforms. The later analysis shows that MPI communication is sometimes factored into the kernels within each timed portion - how do the authors deal with this on the NVIDIA GPU system whereby communication is initiated by the host CPU (as explained previously in the paper around GPU-aware MPI).

*Measuring and comparing different hardware architectures is challenging, because they can have different ways of handling the computations and provide a different set of hardware counters which might measure values in slightly different ways (even when using the same tool). Therefore we explained in Section 4.2.1 what we did to conduct these measurements and make them as compatible as possible. The three sections CPU, GPU and Vector Engine describe this in*

*detail, i.e. using LIKWID, Nsight and ftrace to have comparable results. As described in the GPU part, the actual measurements of the offloaded kernels are considered without additional MPI communication over the compute regions of interest. We did the same for the other two architectures, specifically, the last paragraph of section 4.2.1 describes the relation of actual and measured FLOP/s on vector engines with ftrace and how this is handled in our measurements to be comparable.*

21. This section is a high level discussion on computing architectures again, repeating much of the sentiments from Section 2. The way the argument is presented does not justify the need for the modularisation code, although obviously I'm supportive of modularising the code as described in Figure 2.

*Since this section is the outlook of our paper in the direction of energy efficiency, we refer to some conclusions from Section 2 to make a transition to this topic.*

22. The energy comparison in Figure 8 is interesting here. It would be useful to add the system power draws on Table 2.

*We do not have specific and reliable system power draws for our measurments to add.*

23. The comments around lines 370 are not clear. It is also not clear how remarkable the curve is given the scaling results presented previously, where the compute resource (i.e., energy use) is doubled but yields less that double the performance.

*Assuming that the reviewer is referring to the discussed results shown in Figure 3 and Figure 8, the y axis in Figure 8 shows energy per simulated day and not the absolute performance in simulated days per day. Although increasing the amount of compute resources (i.e. number of GPUs or Vector Engines) results in faster running times – which can also be seen in the x axis in Figure 8 – the energy per simulated day is increasing even more and not scaling in the same way.*

24. The authors again state that the monolithic design is a factor in the performance portabililty. However, the Roofline models show that the codes do roughly equally well on the architectures they are executed on, so I'm not convinved the evidence presented leads to this conclusion specifically.

*In Section 6 we summarize the challenges to sustained exascale performance for the ICON model. We do not state that the monolithic design is an obstacle to single node performance. We just state that modularization should make achieving that goal easier.*